# Grp94 Inhibitor HCP1 Inhibits Human Dermal Fibroblast Senescence

**DOI:** 10.3390/genes13091651

**Published:** 2022-09-14

**Authors:** Xiaoling Cui, Xuxiao Hao, Jie Wen, Shangli Zhang, Baoxiang Zhao, Junying Miao

**Affiliations:** 1Shandong Provincial Key Laboratory of Animal Cells and Developmental Biology, School of Life Science, Shandong University, Qingdao 266237, China; 2School of Stomatology, Shandong First Medical University & Shandong Academy of Medical Sciences, Jinan 250117, China; 3Institute of Organic Chemistry, School of Chemistry and Chemical Engineering, Shandong University, Jinan 250100, China

**Keywords:** senescence, HCP1, AMPK/mTOR signal pathway, lysosomes, mitochondria

## Abstract

Researchers are paying more and more attention to aging, especially skin aging. Therefore, it is urgent to find an effective way to inhibit aging. Here, we report a small chemical molecule, HCP1, that inhibited the senescence of human dermal fibroblasts (HDFs). First, we performed morphological experiment and found that HCP1-treated HDFs were no longer elongated and flat compared to DMSO-treated groups. Next, we found that the number of β-gal positive cells decreased compared to DMSO-treated groups. Through flow cytometry, western blot, and immunofluorescence, we found that HCP1 could inhibit the senescence of HDFs. In the study of the mechanism, we found that HCP1 could regulate the AMPK/mTOR signal pathway through glucose-regulated protein 94 (Grp94). In addition, we found that HCP1 could promote the interaction between Grp94 and lysosomes, which led to an increase in the activity of lysosomes and inhibited the senescence of HDFs. At the same time, we found that HCP1 decreased the concentration of Ca^2+^ in mitochondria, inhibiting the senescence of HCP1. Therefore, we propose that HCP1 is a potential aging-inhibiting compound, and provide a new idea for the development of senescence-inhibiting drugs.

## 1. Introduction

In essence, aging involves the progressive loss of maximum function, stress resistance, metabolic efficiency, and adaptive capability [1,2]. As a result, it is urgent to find mechanisms that contribute to aging and effective ways to delay its onset. Aging of the skin is a complex and multi-factor comprehensive process, which can be divided into intrinsic aging and extrinsic aging depending on the cause of aging [3,4]. Dermis, the second layer of the skin, is a post-mitotic tissue that relies on adaptation and damage repair for homeostasis [5,6,7]. As a major type of cell in the dermis, dermal fibroblasts (HDFs) secrete extracellular matrix, which determines the structure and mechanical properties of the skin [8]. In experimental studies, fibroblast senescence is associated with an irreversible cell proliferation arrest and an increase in the expression of a senescence-associated secretory phenotype (SASP), which results in skin resorption and aging [9,10]. Due to the importance of HDF senescence to the skin, HDFs have become the cells of choice for studying aging.

The problem of aging is one of the most perplexing aspects of human life. There are various anti-aging approaches, including topicals, energy-based procedures, and dermal fillers, that help restore dermal aging’s molecular signature [11]. However, no efficient way to inhibit aging has been found yet. Therefore, it is urgent to find an efficient way to inhibit cellular senescence. Here, we report that a new glucose-regulated protein 94 (Grp94) inhibitor a coumarin pyrazoline derivative 3-(1,5-diphenyl-4,5-dihydro-1H-pyrazol-3-yl)-7-hydroxy-2H-chromen-2-one (HCP1) inhibits the senescence of HDFs [12].

In our previous studies, we found HCP1 inhibited the activity of AMPK by inhibiting the activity of Grp94 [13]. Studies have shown that the AMPK/mTOR signal pathway played a crucial role in senescence [8,14,15]. Simultaneously, AMPK1 was mainly found in mitochondria and lysosomes. In addition, the activity of lysosomes and acidic vacuoles are reduced in senescent cells [16]. Moreover, lysosomes are the main catabolic organelles of cells, helping to remove senescent and dead organelles in cells. Since lysosomes are central hubs of key cellular trafficking, signaling, and metabolic pathways, targeting lysosomes to inhibit cellular senescence has become a widespread research strategy. Senescent cells can be characterized by a number of distinct changes to their homeostatic mechanisms; specifically, mitochondrial biogenesis is exacerbated, mitophagy is decreased, and mitochondrial networks become hyperfused [17]. Therefore, in this article we identified the effect of a Grp94 inhibitor HCP1 on the AMPK/mTOR signal pathway in senescent HDFs.

## 2. Materials and Methods

### 2.1. Cell Culture

HDFswere obtained from Wu Xunwei’s lab (Jinan, China); they were derived from the foreskin tissue of children [18]. HDFs were maintained in DMEM Basic medium (C11995500BT, Gibco, Grand Island, NY, USA) supplemented with 10% (*v*/*v*) bovine calf serum at 37 °C in a 5% CO_2_ atmosphere. After that, cells were seeded into cell culture plates for related experiments. We used population doubling level (PDL) < 10 to characterize healthy cells and PDL > 30 to characterize senescent cells.

### 2.2. Cell Morphology

DMSO-treated healthy cells (PDL < 10) were used as positive control, DMSO-treated senescent cells (PDL > 30) were used as negative control, and senescent cells (PDL > 30) seeded in 6-well plates were treated with different concentrations (0.1, 0.2, and 0.5 μM) of HCP1 for 48 h; the morphological of HDFs were examined under the inverted phase-contrast microscope (Nikon, Tokyo, Japan).

### 2.3. Senescence-Associated β-Galactosidase Assay

HDFs were fixed in 4% tissue cell fixative solution for 15 min, then washed 3 times with PBS. After removing PBS, 1 mL of SA-β-gal staining solution was added to each well for staining for more than 18 h at 37 °C. The senescent HDFs were detected using the inverted phase-contrast microscope (Nikon, Tokyo, Japan). We characterized senescent cells using cells stained blue. Image J counts the number of positive cells.

### 2.4. Flow Cytometry

After the cells were digested into single cells with 0.25% trypsin, they were centrifuged at 1000 rpm for 5 min, the supernatant was discarded, and they were washed twice with 1XPBS. A total of 1 × 10^6^ or 1 × 10^7^ cells was collected for subsequent experiments. The cells were fixed in 1 mL of 70% ethanol at 4 °C overnight. The next day, cells were centrifuged to remove the supernatant and washed three times with 1XPBS, then the nuclei were stained with a staining solution (100 μg/mL propidium iodide (PI) (Biolegend, San Diego, CA, USA), 50 μg/mL RNAase (TIANGEN, Beijing, China). Cells were incubated in staining solution at 4 °C for 15–30 min in the dark. Cells were harvested by flow cytometry (ImageStreamX MarkII, Merck, Billerica, MA, USA). After that, we used IDEAS (IDEAS version 6.0, Amnis, Seattle, WA, USA) to analyze the data.

### 2.5. Immunoprecipitation (IP)

HDFs lysates were pre-cleared with protein A/G agarose beads (P2012, Beyotime, Shanghai, China,) for 1 h at 4 °C. After centrifugation, the supernatant was collected and incubated with specific antibodies or normal corresponding IgG, then with protein A/G beads overnight at 4 °C. The beads were washed with IP buffer 3 times and eluted with 2 × SDS loading buffer. The immunoprecipitated proteins were detected by western blot assay.

### 2.6. Western Blot Analysis

HDFs were lysed in lysis buffer (Beyotime, Beijing, China) containing PMSF (Sigma-Aldrich, St. Louis, MO, USA). Cell lysates were separated by 12% or 9% SDS/PAGE at 4 °C and then transferred to PVDF membrane (Millipore, Billerica, MA, USA). The membrane was blocked with 5% nonfat milk for 1 h at room temperature, incubated with primary antibodies: anti-p21, anti-Collagen I, and anti-HP1γ (Proteintech, Wuhan, China); anti-β-actin (Sigma-Aldrich, St. Louis, MO, USA); anti-p-HP1γ, and anti-Grp94 (Abcam, Cambridge, MA, USA); anti-LAMP1, anti-AMPK, anti-p-AMPK, anti-mTOR, anti-p-mTOR, anti-p-p70S6K, and anti-p70S6K (Cell Signaling Technology, Danvers, MA, USA) at 4 °C overnight and detected with corresponding horseradish peroxidase-conjugated secondary antibody (Jackson ImmunoResearch, West Grove, PA, USA, goat anti-rabbit:13963, goat anti-mouse:130389) at room temperature for 1 h. The membranes were incubated with Immobilon Western Chemiluminescent HRP Substrate for 5 min and then exposed to X-ray film (Kodak, Rochester, NY, USA). The relative protein level was analyzed by imagej software (NIH, Bethesda, MD, USA).

### 2.7. Immunofluorescence Microscopy

We seeded HDFs onto confocal dishes and used HCP1 to treat cells. After treatment, HDFs were fixed in 4% paraformaldehyde for 15 min. After washing three times with 1× PBS, cells were permeated with 0.1–0.2% TritonX-100 for 2 min, and then blocked with donkey serum (1:30 dilution in 0.1 M PBS) for 30 min at room temperature. We discarded the enclosed liquid, incubated cells with primary antibodies: HP1γ (1:100 dilution) overnight at 4 °C, then incubated cells with secondary antibodies for 60 min at 37 °C. Finally, the nuclei were stained with DAPI for 10 min, and a confocal laser scanning microscope (Carl Zeiss, Jena, Germany) was used to detect the fluorescence intensity.

### 2.8. Lysosomal pH Sensing Experiment

HDFs were seeded in confocal dishes (SPL Life Sciences, Pocheon-si, Korea) and cells were processed. Next, 0.5 μM Lysosensor™ Green DND-189 (pH = 5.2) (Invitrogen, Carlsbad, CA, USA) treated cells for 30 min, and 1× PBS washed three times. Confocal laser scanning microscope 700 (Carl Zeiss, Jena, Germany) was used to detect the fluorescence intensity. The excitation wavelength of Lysosensor™ Green DND-189 is 443 nm.

### 2.9. The Staining of Ca^2+^ in Mitochondria

HDFs were treated with DMSO or HCP1 for 48 h. Them, Rhod2 (Thermo Fisher Scientific, MA, USA) of DMSO stock solution (1 mM) was diluted to a final concentration of 2.5 µM in the buffered physiological medium. Addition of the non-ionic detergent Pluronic^®^F-127 aided in the dispersion of the nonpolar AM ester in aqueous media (Rhod2: F127 = 1:1). HDFs are normally incubated with the AM ester for 15–60 min at 20–37 °C. Cells were washed in indicator-free medium to remove any dye that was non-specifically associated with the cell surface, and were then incubated for 12–24 h to allow complete de-esterification of intracellular AM esters and eliminate cytosolic staining, while retaining mitochondrial staining. The fluorescence was detected by a confocal laser scanning microscope 700 (Carl Zeiss, Jena, Germany).

### 2.10. Statistical Analyses

GraphPad Prism software (version 5.0, San Diego, CA, USA) was used to perform statistical analysis. Data were represented as mean ± SEM and analyzed by one-way ANOVA. Pictures were processed with Adobe Photoshop software (version 2017.1.6, Adobe Systems, San Jose, CA, USA). The mean values were derived from at least three independent experiments. Differences at *p* < 0.05 were considered statistically significant.

## 3. Results

### 3.1. HCP1 Inhibits the Senescence of HDFs

To find small chemical molecules that can inhibit dermal fibroblast senescence, we treated senescent HDFs with the small chemical molecule HCP1 and observed whether HCP1 could inhibit the senescence of HDFs. First, we used sulforhodamine B (SRB) to detect the viability in senescent HDFs, and found that HCP1 did not affect the survival of senescent HDFs (Appendix A). Next, DMSO-treated healthy HDFs (PDL = 4) and senescent HDFs (PDL = 31) were used as positive and negative controls; we treated HDFs with HCP1 at different doses (0.1 μM, 0.2 μM, 0.5 μM), and detected the cell morphology. We observed enlarged, flattened cell morphology in the DMSO-treated senescent cell group, which showed features of senescence. However, in the HCP1-treated group, features of cellular senescence gradually disappeared in a concentration-dependent manner (Figure 1a). Next, we examined the effect of HCP1 on senescent HDFs using β-gal staining. Consistently, starting at 0.2 μM, HCP1 reduced the percentage of senescence-associated β-galactosidase-(SA β-gal) positive cells (Figure 1b,c). Taken together, we preliminarily concluded that HCP1 could inhibit the senescence of HDFs.

### 3.2. HCP1 Regulated Cell Cycle in Senescent HDFs

Studies have shown that senescent cells exhibit changes in growth kinetics that arrest growth and exit the cell cycle [19]. In order to further verify that HCP1 inhibits HDFs senescence, we treated HDFs with DMSO or HCP1 for 48 h, and performed flow cytometry to detect the cell cycle of HDFs. We found about 88.02% senescent HDFs remained in the G0/G1 phase, while in the HCP1-treatment group, the percentage of cells staying at G0/G1 phase decreased to about 83.73% (Figure 2a).

Then, in order to further check the inhibitory effect of HCP1 on the HDFs cell cycle, we conducted Western blotting to detect the protein level of P21, which was an important protein that regulated the cell cycle. Data revealed that HCP1 reduced the protein level of p21 in senescent HDFs in a dose-dependent manner (Figure 2b,c). In summary, all these data demonstrated that treating senescent HDFs with HCP1 could prevent them from exiting the cell cycle.

### 3.3. HCP1 Improved the Protein Level of Type I Collagen and Senescence-Associated Heterochromatin Foci (SAHF)

Studies have shown that SASP displays pleiotropic effects [20]. However, type I collagen, as the main component of the extracellular matrix, expresses higher amounts in HDFs of young individuals than in fibroblasts from older donors [21]. We investigated the effect of HCP1 on senescence using healthy HDFs as a positive control. We treated senescent HDFs with different dose-HCP1 for 48 h and analyzed the protein level of type I collagen. Western blotting showed that HCP1 promoted the elevated protein level of type I collagen in dose-dependent manner, and the increase is most obvious at 0.5 μM (Figure 3a,b).

SAHF often appear in the nucleus of senescent cells, which are composed of facultative heterochromatin and deepened staining [22]. The formation of SAHF will cause the silencing of some pro-proliferation genes. Some components of SAHF include heterochromatin protein 1 (HP1) and lysine 9 di- or tri-methylated histone H3 (H3K9Me2/3) [23]. We hypothesized that HCP1 reduced the formation of SAHF. We conducted immunofluorescence and found that in the case of DAPI staining, a large number of deepened spotted areas appeared in the nucleus of senescent cells, and a large number of SAHF components-HP1-γ appeared on these deepened areas. Data showed that there was a large amount of SAHF in senescent HDFs, while 0.5 μM HCP1 could significantly reduce SAHF in senescent cells (Figure 3c,d).

It has been shown that HP1γ, which plays an important role in heterochromatin formation, is phosphorylated to enhance HP1γ’s capacity to bind to H3K9me3, resulting in heterochromatin formation and suppression of proliferative genes, such as CCNA2 and PCNA [24]. To further investigate the effect of HCP1 on senescent HDFs, Western blotting was performed on phosphorylation of HP1γ. The results demonstrated that HCP1 treatment of senescent HDFs can reduce the phosphorylation level of HP1γ (Figure 3e,f).

### 3.4. HCP1 Failed to Function in Nucleus during the Suppression of HDFs Senescence

To elucidate the reason why HCP1 inhibits HDFs senescence, we investigated the signal pathway affected by HCP1. In previous reports, we have found that HCP1 bound Grp94 to inhibit the activity of Grp94 [25]. We examined whether Grp94 could interact with HP1γ to regulate the effect of HP1γ on senescence. First, we used immunofluorescence to detect the colocalization of Grp94 with HP1γ. Disappointingly, we found that Grp94 failed to co-localize with HP1γ (Figure 4a). We also used co-immunoprecipitation to further verify the above experimental results, and the conclusions were consistent (Figure 4b). This also implied that HCP1 failed to function in the nucleus, whereas HCP1 may play a role in the cytoplasm.

### 3.5. HCP1 Regulated the AMPK/mTOR Signal Pathway

In the above experiments, we hypothesized that HCP1 inhibited the senescence of HDFs by playing a role in cytoplasm. Studies have shown that the AMPK/mTOR signal pathway is critical for cellular senescence. In our previous study, we found that HCP1 could inhibit the activity of AMPK in HUVECs, which is related to the inhibition of Grp94 activity by HCP1 [13], but whether HCP1 regulated the AMPK/mTOR signal pathway in HDFs was unclear. To further investigate whether HCP1 inhibits HDFs senescence through the Grp94/AMPK/mTOR signal pathway, we examined the effect of HCP1 on the AMPK/mTOR signal pathway. First, we treated senescent HDFs with HCP1 for different amounts of time, and Western blotting was conducted to detect the phosphorylation of AMPK. The results showed that HCP1 reduced the phosphorylation of AMPK in a time-dependent manner in senescent HDFs (Figure 5a,b). Next, we detected the impact of HCP1 on mTOR signal pathway in the presence of Rapa. Consistently, we found that HCP1 could activate the mTOR signal pathway, but in the existence of Rapa, HCP1 failed to work (Figure 5c–e). However, AICAR, an activator of AMPK, surely suppressed HCP1-decreased the phosphorylation of AMPK (Figure 5f,g). Taken together, these results revealed that HCP1 could regulate the AMPK/mTOR signal pathway during HCP1 inhibiting senescence of HDFs.

### 3.6. HCP1 Protected and Enhanced the Activity of Lysosome via Protecting the V0 Proton Channel of v-ATPase

Current research proved that AMPK1 was distributed in a variety of organelles, mainly distributed in lysosomes and mitochondria. In order to further study the signal pathway affected by HCP1 during the senescence of HDFs, we analyzed the effect of HCP1 on lysosome and mitochondrial [26,27]. To examine whether HCP1 inhibits the senescence of HDFs through lysosomes, we detected the colocalization of Grp94 and LAMP1, the result showed that HCP1 could promote the co-localization of Grp94 and LAMP1, which also confirms that HCP1 could play a role in lysosomes during HCP1 inhibiting the senescence of HDFs (Figure 6a,b). Simultaneously, we also analyzed the protein level of LAMP1, which included the biomarkers of lysosomal function. Immunofluorescence staining showed a reduced level in DMSO-treated senescent cells. Treating senescent HDFs with HCP1 significantly promoted the increased protein level of LAMP1, which restored the lysosomal activity (Figure 6c,d). Consistently, the result was also verified by Western blotting (Figure 6e,f). Together, these data suggested that HCP1 might have a protective function on damaged lysosomes.

Lysosomal acidity is closely related to lysosomal activity. Wang et al. demonstrated a decrease in lysosomal activity with age, as well as a decrease in acidic vacuoles [28]. We hypothesized that lysosomal activity is altered during HCP1 inhibition of HDFs senescence. To test this hypothesis, we first incubated senescent HDFs with HCP1 for 0, 12, 24, and 48 h, and then used Lysosensor™ Green DND-189 to quantify the H+ concentration in the lysosome. We found that HCP1 can significantly increase the H+ concentration in lysosomes in a time-dependent manner, indicating that HCP1 can increase the H+ concentration in lysosomes of senescent cells, thereby increasing lysosomal activity and protecting lysosomes (Figure 6g,h).

The vacuolar ATPase is able to pump protons (H) into the lysosome, thereby lowering the pH in the lysosome and activating the activity of the enzyme in the lysosome. However, it is unclear whether the decrease in lysosomal pH in HDF is the result of protection of v-ATPase. The inhibition site of Baf-A1 localizes to the V0 proton channel and requires residues of subunit C. Next, we explored the targeting sites of HCP1 with or without Baf-A1 and/or HCP1, and detected the fluorescence intensity by Lysosensor™ Green DND-189 staining. As expected, HCP1 failed to increase the H+ concentration in lysosomes in the presence of Baf-A1, which indicated that HCP1 raised the H+ concentration in lysosomes via protecting v-ATPase (Figure 6i–l).

### 3.7. HCP1 Reduced the Concentration of Ca^2+^ in Mitochondria

Next, in order to explore whether HCP1 affected mitochondria, we examined the concentration of Ca^2+^ in mitochondria in the presence of HCP1. Most calcium in the ER is stored bound to proteins, and GRP94 is one of a few major luminal calcium-binding proteins [29]. Studies have shown that ER calcium released through ITPR2 channels leads to mitochondrial calcium accumulation and senescence [30]. In our study, we found that HCP1 reduced the concentration of mitochondrial Ca^2+^ in a dose- and time-dependent manner (Figure 7a–d). Taken together, this result demonstrated that HCP1 regulated decreased mitochondrial calcium levels during inhibiting the senescence of HDFs.

## 4. Discussion

The skin consists of three parts: epidermis, dermis, and subcutaneous tissue. The epidermis is a multi-layered epithelium composed of keratinocytes. With age, the epidermal layer becomes thinner and damaged, which can compromise the skin’s epidermal barrier [31,32]. The dermis is mainly composed of dermal fibroblasts and extracellular matrix [33]. Aging is accompanied by remodeling and degradation of the extracellular matrix, leading to changes in skin structure and skin aging [34]. However, the number of older persons in the world is increasing. Therefore, finding an effective way to inhibit aging has become a major goal of current research. In our study, we discovered that a small chemical molecule HCP1 could effectively inhibit HDFs senescence. In our study, we discovered that an inhibitor of Grp94, HCP1, could effectively inhibit HDFs senescence. We demonstrated that HCP1 is able to reduce the percentage of β-gal positive cells and prevent cells from exiting the cell cycle. At the same time, we found that HCP1 can alter SAHF and SASP, which greatly affects the function of senescent cells.

Existing studies mostly focus on the senescence of HDFs induced by ultraviolet rays [35]. With the wide application of sunscreens and the increasing improvement of sunscreen equipment, the senescence of dermal fibroblasts induced by ultraviolet rays can be largely avoided [36,37]. At the same time, increasing attention is being paid to the senescence of fibroblasts associated with aging, termed replicative senescence [38,39]. Replicative senescence is a phenomenon in which telomeres are shortened due to the continuous division of cells, which induces cell senescence and contributes to organismal aging [40,41]. However, there is a lack of effective ways to prevent the changes in skin state caused by replicative senescence. Here, we used dermal fibroblasts with PDL > 30 to mimic senescent cells and found that the small chemical molecule HCP1 can improve the senescence of dermal fibroblasts in vitro, which provides a new tool and entry point to study the molecular mechanism of natural HDFs senescence.

Senescence studies models based on the nature of stimulation can be divided into telomere-dependent senescence and telomere-independent senescence, termed replicative senescence and stress-induced premature senescence, respectively [42,43]. The occurrence of replicative senescence is mainly related to telomere shortening. However, in addition to replicative senescence, a range of non-telomere stimuli can lead to premature cellular senescence independent of telomere length [44,45]. These stimuli are mainly caused by internal or external, physical or chemical, acute or chronic substances, such as oxidative stress, genotoxicity, oncogenes, etc. [46,47,48,49]. At present, the common features exhibited by cellular senescence include DNA damage, induction of p53/p21 signaling pathway, and high activity of senescence-related β-galactosidase, SAHF, SASP, etc. However, different cells may exhibit one or more of these characteristics based on different stimuli [50]. As a result, scientists are hampered in studying and treating aging due to the lack of stable and universally measurable traits. In our study, we used HCP1 to investigate the aging traits and signaling pathways exhibited by dermal fibroblasts during replicative aging, providing some insights into skin aging research.

In our previous study, we found that HCP1 inhibited AMPK/mTOR signal pathway through inhibiting the activity of Grp94. AMPK/mTOR signal pathway played an important role in cell senescence. However, it was unclear whether HCP1 regulated HDFs senescence through AMPK/mTOR signal pathway. Here, we found that the activity of AMPK was inhibited and mTOR was activated during HCP1 inhibition of HDFs senescence. More and more studies have shown that AMPK/mTOR affects the function of mitochondria and lysosome [51].

Some studies have suggested that the lysosomal dysfunction may lead to cellular senescence, and ultimately, progressive neurodegenerative disease [52,53]. There is increasing evidence that lysosomal acidification affects the activity of various hydrolases within the lysosome, as well as various additional signal functions of the lysosome, especially the regulation of nutrient sensing and nutrient homeostasis [26,54]. The proper coordinated function of multiple ion channels is involved in the regulation of lysosomal acidification, most notably the vacuolar ATPase, which is responsible for pumping protons (H) into the lysosome and lowering the intraluminal pH to the acidic range required to activate dozens of hydrolases. Interestingly, we found that HCP1 promoted the interaction between Grp94 and lysosome, which had not been reported before. This study suggests that Grp94 may regulate cellular senescence by regulating lysosomal function.

Emerging evidence has pinpointed mitochondria as one of the key modulators in the development of the senescence phenotype, particularly the pro-inflammatory senescence associated secretory phenotype (SASP). Simultaneously, Grp94 has been identified as one of a few major luminal calcium-binding proteins that regulated calcium homeostasis in ER [29]. However, calcium ions in the endoplasmic reticulum can enter the mitochondria through the MCU for mitochondrial calcium accumulation, which ultimately leads to a subsequent decrease in mitochondrial membrane potential, reactive oxygen species accumulation, and senescence [30]. Here, we found that in the process of HCP1 inhibiting HDFs senescence, HCP1 reduced the level of Ca^2+^ in mitochondria. Therefore, our study demonstrated that HCP1 could affect the functions of lysosome and mitochondria in senescent HDFs.

## 5. Conclusions

In summary, we discovered that a new inhibitor of Grp94, HCP1, inhibited replicative senescence in HDFs. At the same time, in the process of HCP1 inhibiting aging, we found that HCP1 inhibited the activity of AMPK through inhibiting the activity of Grp94, which leads to inhibit the senescence of HDFs. On the other hand, we found that HCP1 could affect the functions of lysosome and mitochondria through Grp94 (Figure 8). This study provides a new tool and opinion for the study of replicative senescence, as well as the mechanism study for Grp94 regulating senescence.

## Figures and Tables

**Figure 1 genes-13-01651-f001:**
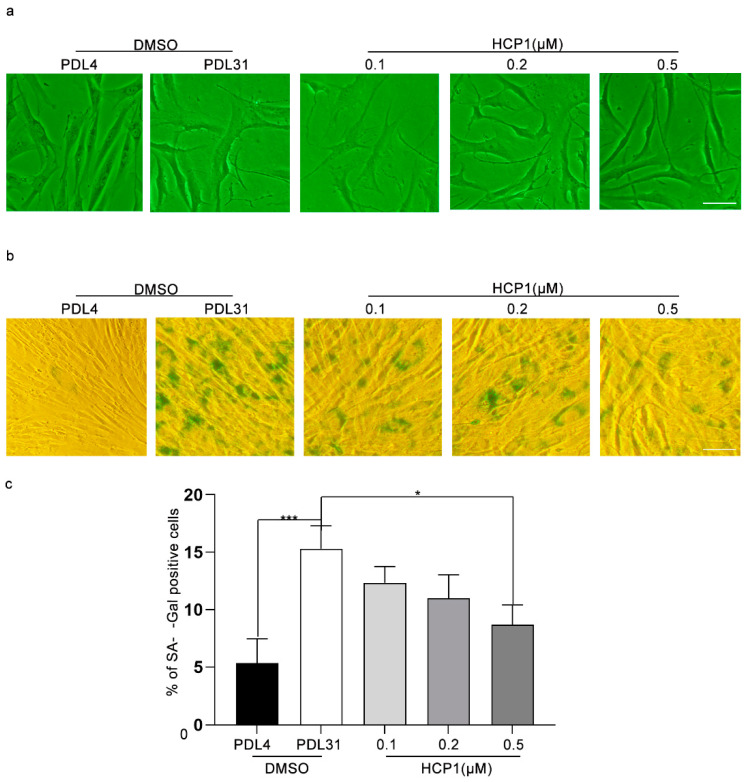
The effect of HCP1 on senescence HDFs. (**a**) DMSO or HCP1 (0.1, 0.2, 0.5 μM) treated senescent HDFs (PDL = 31) for 48 h, the morphological change of senescent cells was observed by inverted phase contrast microscope (Eclipse TS-100; Nikon, Tokyo, Japan); (**b**,**c**) Senescent HDFs were treated with DMSO or HCP1 (0.1, 0.2 and 0.5 μM) for 48 h. SA-β-gal was conducted to stain the senescent HDFs, and inverted phase contrast microscope was used to observed senescent HDFs. Percentage of SA-β-gal-positive cells was counted. Healthy cells (PDL = 4) as positive control. Scale bar: 100 μm. Data are presented as means ± SEM, * *p* < 0.05, *** *p* < 0.001, *n* = 3.

**Figure 2 genes-13-01651-f002:**
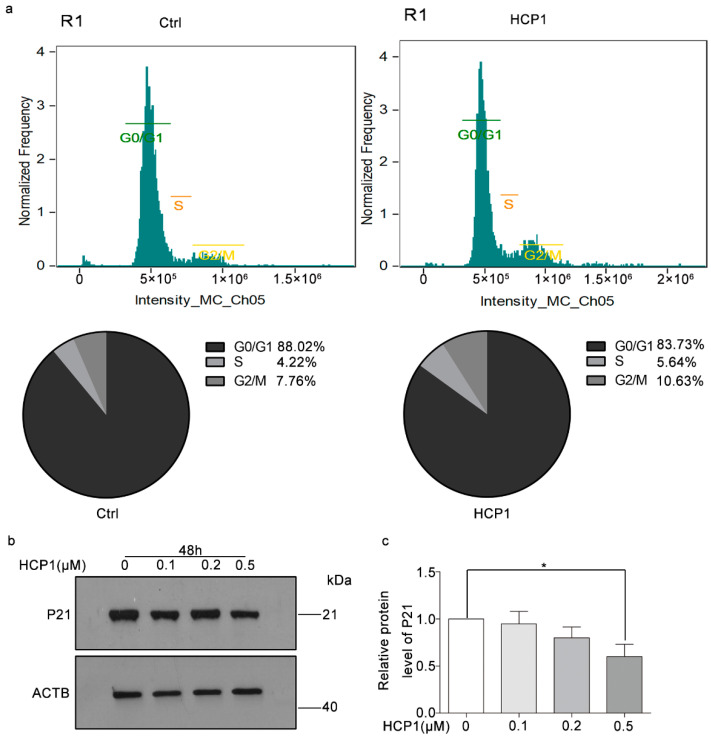
HCP1 regulated cell cycle in senescent HDFs. (**a**) Senescent HDFs were treated with DMSO or HCP1 (0.5 μM) for 48 h, the proliferation of senescent HCP1 was analyzed by flow cytometry. (**b**,**c**) HCP1 (0, 0.1, 0.2, 0.5 μM) treated senescent HDFs for 48 h, Western blot was used to detect the protein level of P21. Data are presented as means ± SEM, * *p* < 0.05, *n* = 3.

**Figure 3 genes-13-01651-f003:**
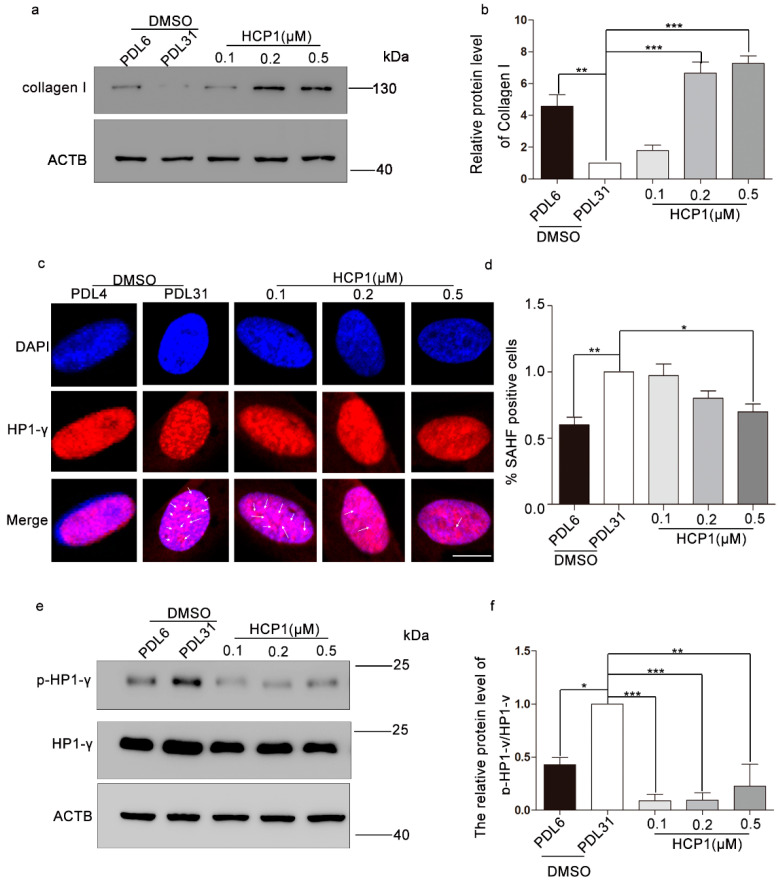
Effects of HCP1 on Collagen I and SAHF in senescent HDFs. (**a**,**b**) After senescent HDFs (PDL = 31) were treated with DMSO or HCP1 (0.1, 0.2, and 0.5 μM) for 48 h, the protein level of Collagen I was detected by Western blot. (**c**,**d**) After DMSO or HCP1 (0.1, 0.2, 0.5 μM) treated senescent cells for 48 h, immunofluorescence experiments were performed with HP1-γ antibody and DAPI to detect the aggregation of HP1-γ and statistical analysis. White arrows indicate deepened spotted areas. Scale bar: 20 μm (**e**,**f**) Senescent HDFs treated with DMSO or HCP1 (0.1, 0.2, and 0.5 μM) for 48 h. The phosphorylation level of HP1γ was detected by Western blot. DMSO treated-healthy HDFs as positive controls. Data are presented as means ± SEM, * *p* < 0.05, ** *p* < 0.01, *** *p* < 0.001, *n* = 3.

**Figure 4 genes-13-01651-f004:**
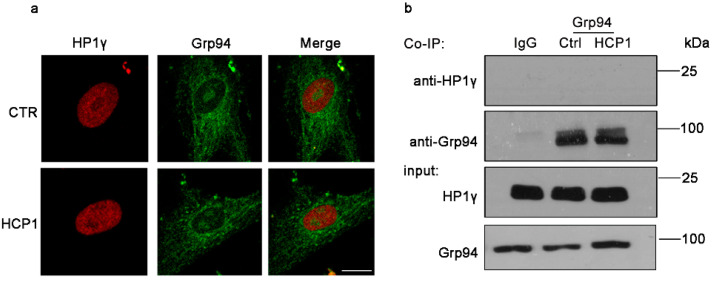
Interaction of HP1γ and Grp94. After senescent HDFs (PDL > 30) were treated with 0.5 μM HCP1 for 48 h, Immunofluorescence (**a**) and Co-IP (**b**) were performed to detect the interaction of HP1γ and Grp94. Scale bar: 20 μm.

**Figure 5 genes-13-01651-f005:**
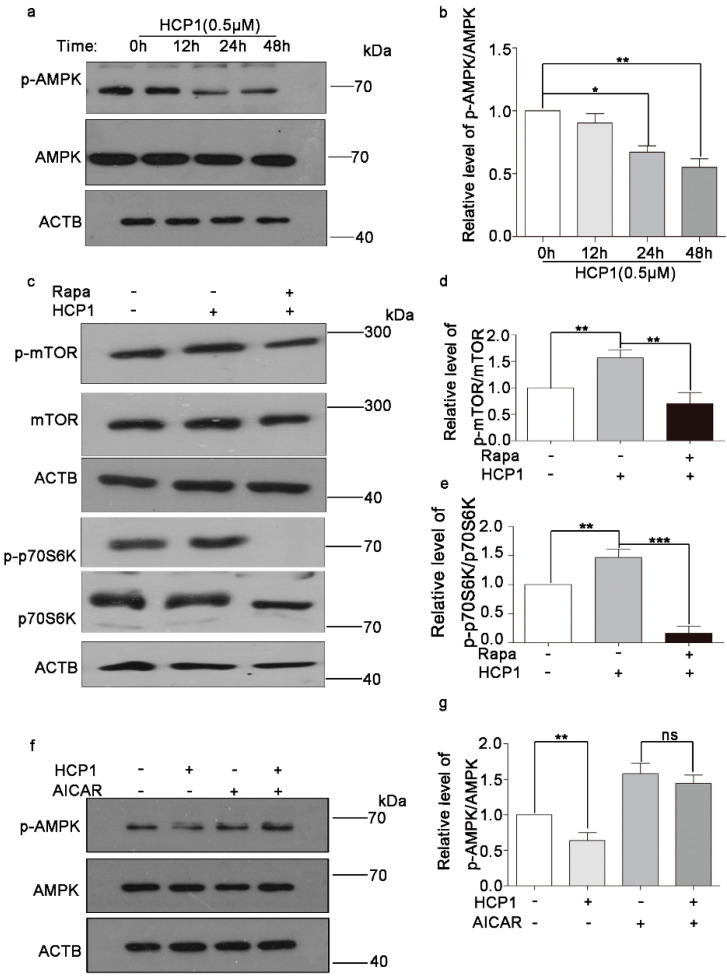
Effects of HCP1 on mTOR/AMPK signal pathway. (**a**,**b**) HDFs (PDL > 30) were treated with 0.5 μM HCP1 for 0, 12, 24, 48 h, Western blotting was used to detected the protein level of p-AMPK or AMPK. (**c**–**e**) HDFs were treated with Rapa or HCP1, Western blotting was used to detected the protein level of the phosphorylation of mTOR and p70S6K. (**f**,**g**) Western blot analysis of phosphorylation of AMPKα (p-AMPKα) and AMPKα in HDFs (PDL > 30) treated with HCP1 or AICAR at indicated concentrations for 48 h. Data are presented as means ± SEM, * *p* < 0.05, ** *p* < 0.01, *** *p* < 0.001, *n* = 3.

**Figure 6 genes-13-01651-f006:**
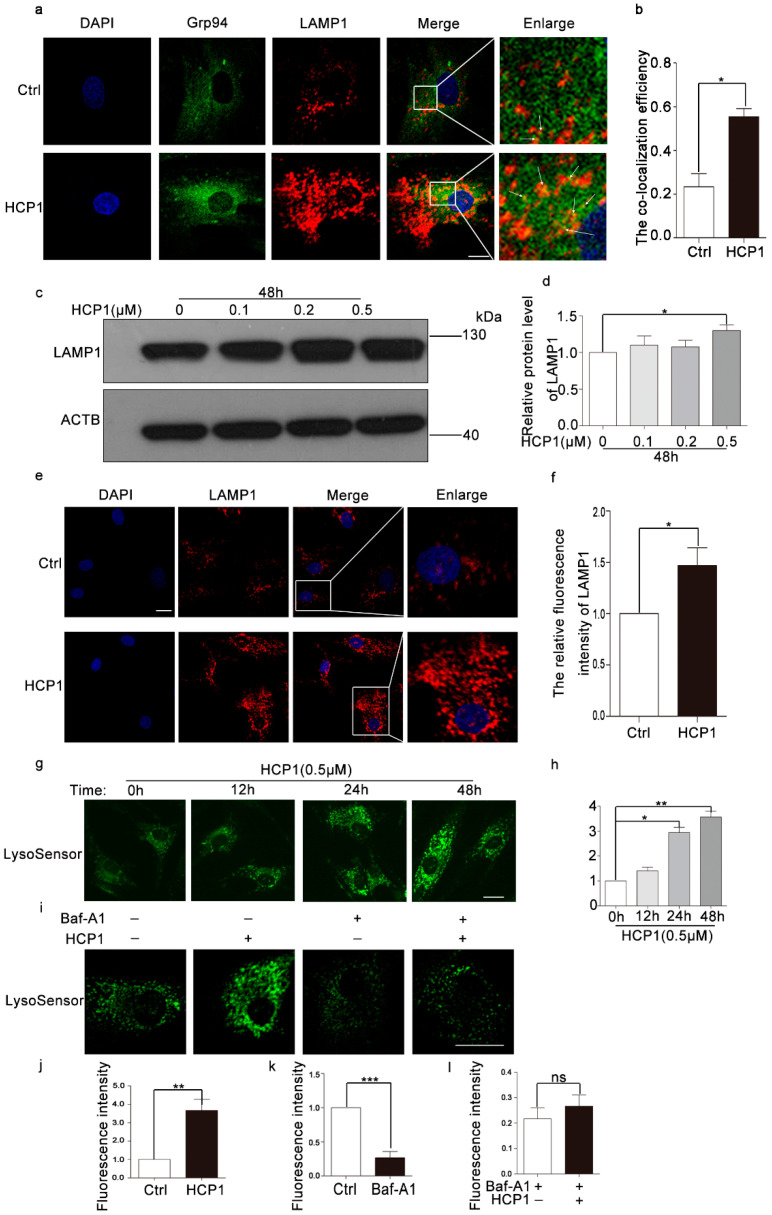
HCP1 up-regulated the activity of lysosome by activating v-ATPase. (**a**,**b**) HDFs were treated with DMSO or HCP1 (0.5 μM) for 48 h, Immunofluorescence analysis the interaction of Grp94 and LAMP1. White arrows indicate colocalized regions. (**c**–**f**) DMSO or HCP1 (0.1, 0.2, 0.5 μM) treated senescent HDFs for 48 h, Western blot and Immunofluorescence analysis of the protein levels of LAMP1. (**g**,**h**) After HCP1 (0.5 μM) treated senescent cells for 12 h, 24 h, 48 h, the lysosomal probe Lysosensor Green DND-189 was used to detect the H+ level in the lysosome. (**i**–**l**) HCP1 (0.5 μM) and/or Baf-A1 (20 nM) treated senescent HDFs for 48 h, a confocal laser scanning microscope 700 was used to detect the fluorescent intensity of Lysosensor Green DND-189. Scale bar: 20 μm. Data are presented as means ± SEM, * *p* < 0.05, ** *p* < 0.01, *** *p* < 0.001, *n* = 3.

**Figure 7 genes-13-01651-f007:**
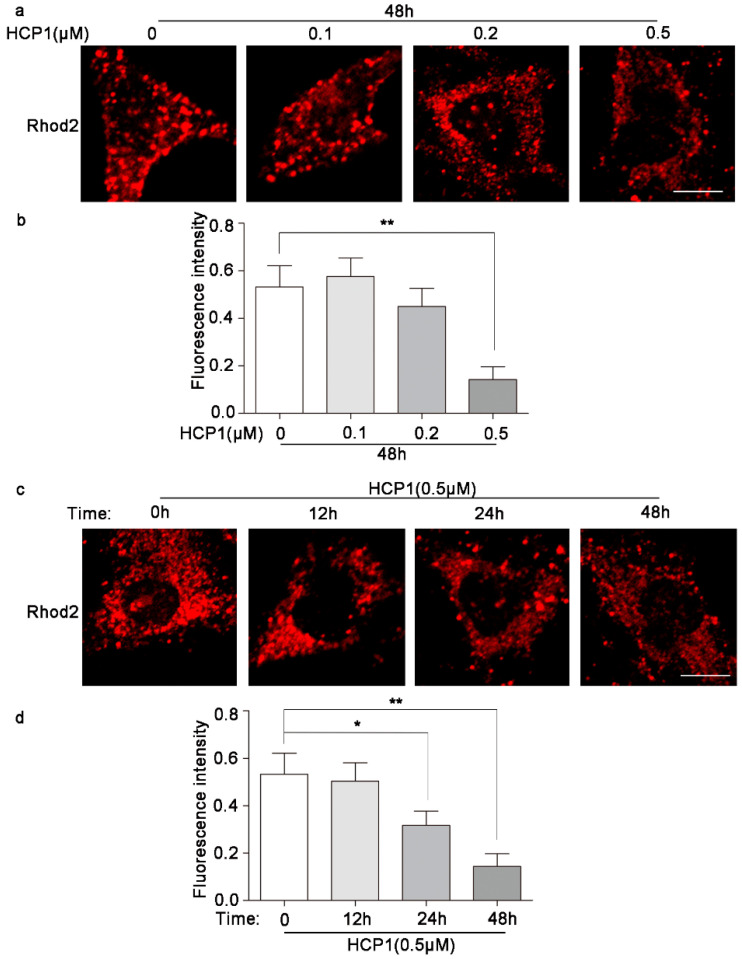
HCP1 reduced the level of Ca^2+^ in mitochondria. (**a**–**d**) HDFs were treated with DMSO or HCP1 (0.1, 0.2, 0.5 μM) at indicated concentrations for 0 h, 12 h, 24 h, and 48 h, Rhod2 was used to detect the level of Ca^2+^ in mitochondria. Scale bar: 20 μm. Data are presented as means ± SEM, * *p* < 0.05, ** *p* < 0.01, *n* = 3.

**Figure 8 genes-13-01651-f008:**
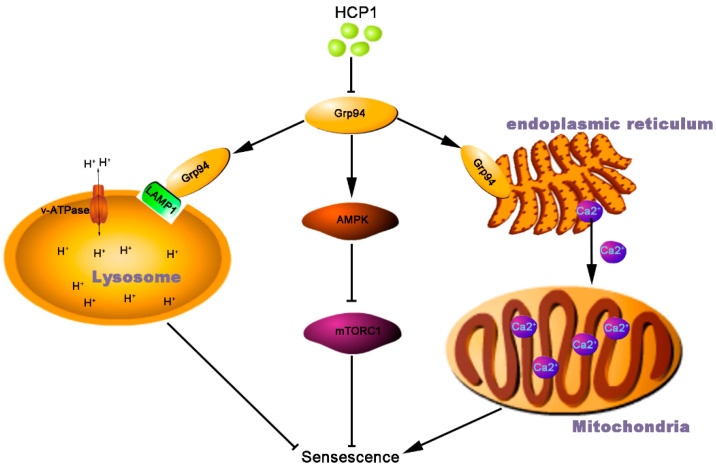
Mechanism of HCP1 inhibiting HDF senescence. On the one hand, HCP1 can inhibit the activity of AMPK through Grp94, thereby regulating the senescence of HDFs through the AMPK/mTOR signal pathway. On the other hand, HCP1 can protect lysosomes and regulate the senescence of HDFs by promoting the interaction between Grp94 and LAMP1. In addition, HCP1 can also promote the increased concentration of Ca^2+^ in mitochondria and inhibit HDFs senescence.

## Data Availability

All relevant data are contained within the article. The data that support the findings of this study are available upon request from the corresponding author.

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
