# Peer review of "Grp94 Inhibitor HCP1 Inhibits Human Dermal Fibroblast Senescence"

_genes, 2022, doi:10.3390/genes13091651_

Round 1
Reviewer 1 Report
In this manuscript, Cui et al explored the effect of Grp94 inhibitor HCP1 on senescence in HDFs. The authors have conducted experiments focusing on how HCP1 could affect the aging of HDFs, and finally they’ve concluded that HCP1 may manipulate the cellular senescence network through multiple aspects such as cell cycle, SHAF, mTOR/AMPK signaling, activity of lysosome and mitochondrion, etc. Overall, this study has provided interesting conception and valuable insights into the regulation of HDFs senescence. However, some major points need to be clarified, and more experimental evidence may also need to be provided.
1 What about the cytotoxicity of HCP1 in HDFs? Please provide the results showing the cell viability after treating with 0, 0.1, 0.2, 0.5μM of HCP1.
2 In Fig 2a, the effect of HCP1 on cell cycle was quite slight. If the cells are resistant to HCP1, you may need to try higher dose in order to get more obvious results.
3 Is there a typo in line 248? It seems like you used HCP1 in Fig 5f, rather than “ZBM-H”.
4 Please reconfirm the western blot for p-p70S6K in Fig 5c. There is no expression of p-p70S6K with Rapa+HCP1 treatment, while the calculation from Fig 5e just showed the significant drop of p-p70S6K expression, not totally gone.
5 The conclusion for Fig 6a-d is not quite convincing. First, I don’t see the obvious co-localization of Grp94 and LAMP1 in Fig 6a, even both of them are mainly detected in cytoplasm. Second, it is hard to say that HCP1 significantly increases the level of LAMP1 from Fig 6c-d. The western blot and its calculation only showed a slight increase (less than 1.3-fold). Actually, immunofluorescence in Fig 6a showed more obvious induction of LAMP1 with the treatment of HCP1.
Minor points:
1 Some inconsistent expression can be found occasionally, for example “aging” and “ageing”.
2 In line 150, please provide more information about the library and screen methods used in this study. If you follow the former publications, please cite them.
Reviewer 2 Report
Your manuscript fits the scope of GENES; it is fairly presented and written; Some major and minor points are, however, to be considered before publication:
Line 33. Please be consistent with basic terminology spelling: “aging” or “ageing.” Choose one style.
Line 35. The dermis is the second layer of the skin. Below dermis, we have hypodermis (https://doi.org/10.3390/ijms21155281)
Line 63. Please, include the original name of the cell line, possible gene modification (if it was made), area of the skin cells that were extracted, and sex and age of the donor of the cells.
Line 88. Please, use appropriate units for the power representation.
Line 90. What brand of flow cytometer did you use, and what software? Please include.
Line 112. Please include in your supplements non-cropped and no photoshopped photos of membranes. And add a link or comment where readers can find it.
Line 158. Figure 1. Please add a magnification scale on photos. Please include. Is 0 uM equal to DMSO-control?
Please mark on the figure control (DMSO) and treatments HCP1. It is essential to include healthy, not senescent cells for comparison with senescent cells.
Line 152. Compared to “DMSO” (0 uM), 0.5 uM HCP1 treated HDF looks more elongated. This is confusing for a reader. (Figure 1a). 0 uM demonstrates a feature of senescent fibroblasts compared to 0.5 uM.
Line 164. Please, exclude unnecessary information like ***p < 0.001; you did not show it in your figure 1c.
Line 203. Please, include untreated healthy cells for comparison and indicate DMSO control. Additionally, use arrows to display SAHF in the figures. SAHF in 0.5 uM HCP1 treated fibroblasts did not look decreased.
Line 217. Please, remove unnecessary information such as **p < 0.01, ***p < 0.001. It is absent in your figures.
Line 250. Please, be consistent with spaces, and remove unnecessary.
Line 317. Remove unnecessary information, ***p < 0.001.
Line 343. “in vitro” must be italicized.
In your discussion, I suggest including a paragraph and/or table with other popular models of in vitro senescence studies models (e.g., stress H2O2 model, UV, hypoxia, antibiotics, etc.)
I suggest you add the following references:
Gerasymchuk, M.; Robinson, G.I.; Kovalchuk, O.; Kovalchuk, I. Modeling of the Senescence-Associated Phenotype in Human Skin Fibroblasts. Int. J. Mol. Sci. 2022, 23, 7124. https://doi.org/10.3390/ijms23137124
Boraldi, F.; Annovi, G.; Tiozzo, R.; Sommer, P.; Quaglino, D. Comparison of ex vivo and In Vitro human fibroblast ageing models. Mech. Ageing Dev. 2010, 131, 625–635
Bertschmann, J.; Thalappilly, S.; Riabowol, K. The ING1a model of rapid cell senescence. Mech. Ageing Dev. 2019, 177, 109–117.
Campisi, J.; d’Adda di Fagagna, F. Cellular senescence: When bad things happen to good cells. Nat. Rev. Mol. Cell Biol. 2007, 8, 729–740.
Ott, C.; Jung, T.; Grune, T.; Höhn, A. SIPS as a model to study age-related changes in proteolysis and aggregate formation. Mech. Ageing Dev. 2017, 170, 72–81.
Gerasymchuk, M.; Robinson, G.I.; Kovalchuk, O.; Kovalchuk, I. The Effects of Nutrient Signaling Regulators in Combination with Phytocannabinoids on the Senescence-Associated Phenotype in Human Dermal Fibroblasts. Int. J. Mol. Sci. 2022, 23, 8804. https://doi.org/10.3390/ijms23158804
Round 2
Reviewer 1 Report
I appreciate the authors' effort to improve the manuscript, which has been revised substantially and all my former concerns have been addressed.